# New Mechanistic Insights on Carbon Nanotubes’ Nanotoxicity Using Isolated Submitochondrial Particles, Molecular Docking, and Nano-QSTR Approaches

**DOI:** 10.3390/biology10030171

**Published:** 2021-02-25

**Authors:** Michael González-Durruthy, Riccardo Concu, Juan M. Ruso, M. Natália D. S. Cordeiro

**Affiliations:** 1Department of Chemistry and Biochemistry, LAQV@REQUIMTE, Faculty of Sciences, University of Porto, 4169-007 Porto, Portugal; ncordeir@fc.up.pt; 2Soft Matter and Molecular Biophysics Group, Department of Applied Physics, University of Santiago de Compostela, 15782 Santiago de Compostela, Spain; juanm.ruso@usc.es

**Keywords:** mitochondria, F0F1-ATPase, carbon nanotubes, computational nanotoxicology

## Abstract

**Simple Summary:**

Carbon nanotubes are revolutionary materials with applications in a lot of different areas. However, there is a rising concern regarding unlikely toxicity effects these materials may trigger. Due to this, the main aim of this paper is to develop a comprehensive approach to study toxicity effect of carbon nanotubes on the mitochondria F0F1-ATPase. We have employed a combination of experimental and computational study. In so doing, we have combined in vitro inhibition responses in submitochondrial particles with docking elastic network models, fractal surface analysis, and Nano-quantitative structure toxicity relationship models (Nano-QSTR models). Results show that this method may be used for the fast prediction of the nanotoxicity induced by single walled carbon nanotubes (SWCNT), avoiding time- and money-consuming techniques, and may open new avenues toward to the better understanding and prediction of new nanotoxicity mechanisms.

**Abstract:**

Single-walled carbon nanotubes can induce mitochondrial F0F1-ATPase nanotoxicity through inhibition. To completely characterize the mechanistic effect triggering the toxicity, we have developed a new approach based on the combination of experimental and computational study, since the use of only one or few techniques may not fully describe the phenomena. To this end, the in vitro inhibition responses in submitochondrial particles (SMP) was combined with docking, elastic network models, fractal surface analysis, and Nano-QSTR models. In vitro studies suggest that inhibition responses in SMP of F0F1-ATPase enzyme were strongly dependent on the concentration assay (from 3 to 5 µg/mL) for both pristine and COOH single-walled carbon nanotubes types (SWCNT). Besides, both SWCNTs show an interaction inhibition pattern mimicking the oligomycin A (the specific mitochondria F0F1-ATPase inhibitor blocking the c-ring F0 subunit). Performed docking studies denote the best crystallography binding pose obtained for the docking complexes based on the free energy of binding (FEB) fit well with the in vitro evidence from the thermodynamics point of view, following an affinity order such as: FEB (oligomycin A/F0-ATPase complex) = −9.8 kcal/mol > FEB (SWCNT-COOH/F0-ATPase complex) = −6.8 kcal/mol ~ FEB (SWCNT-pristine complex) = −5.9 kcal/mol, with predominance of van der Waals hydrophobic nano-interactions with key F0-ATPase binding site residues (Phe 55 and Phe 64). Elastic network models and fractal surface analysis were performed to study conformational perturbations induced by SWCNT. Our results suggest that interaction may be triggering abnormal allosteric responses and signals propagation in the inter-residue network, which could affect the substrate recognition ligand geometrical specificity of the F0F1-ATPase enzyme in order (SWCNT-pristine > SWCNT-COOH). In addition, Nano-QSTR models have been developed to predict toxicity induced by both SWCNTs, using results of in vitro and docking studies. Results show that this method may be used for the fast prediction of the nanotoxicity induced by SWCNT, avoiding time- and money-consuming techniques. Overall, the obtained results may open new avenues toward to the better understanding and prediction of new nanotoxicity mechanisms, rational drug design-based nanotechnology, and potential biomedical application in precision nanomedicine.

## 1. Introduction

The coupled mechanical co-rotating between the γ and ε subunits that form the mitochondrial F1-ATP synthase (complex V) favors the H^+^ protons flux necessary for ATP synthesis in all eukaryotic cells [1,2]. This bioenergetic process involves several synchronized conformational changes which are critical for the survival or death of the cells [1]. In this regard, a few years ago, it was shown that under pathological conditions like chronic diseases such as cancer, Alzheimer’s disease, Parkinson’s disease, and mitochondrial encephalopathy, lactic acidosis (MELAS) syndrome, several toxic events, including nanotoxicity induced by single walled carbon nanotubes (SWCNT), may trigger F0F1ATPase dysfunction [3,4]. As a consequence, the ATP cellular reserves are abruptly consumed by a reverse biochemical reaction which paradoxically hydrolyses significant amounts of ATP, compromising the cellular homeostasis and viability [3,5,6]. Several chemical agents (including carbon nanoparticles) have shown a high affinity/selectivity by the bioenergetic mechanisms based on ATP hydrolysis, particularly nanoparticle-based single-walled carbon nanotubes (SWCNTs), which have been studied by their selective nanotoxicity effects on mitochondria (mitotropic behavior) [7,8,9,10].

To the best of our knowledge, the toxicological modulation of mitochondrial ATP bioenergetic mechanisms released by the exposure with SWCNT-pristine and oxidized-SWCNT (SWCNT-COOH) have been insufficiently characterized in order to explain the mitochondrial nanotoxicity induced by SWCNT. On the other hand, this mechanistic knowledge could be very useful to implement strategies on the named “precision mitochondrial nanomedicine” to improve selectivity for the treatment of brain, cardiac diseases, and cancer using the mitotropic behavior of SWCNT to address active pharmacological principles as new targeting of the mitochondrial F0F1-ATPase [9,10,11,12,13,14]. In this context, we hypothesize that SWCNT-pristine could act by mimicking the pharmacodynamic behavior of the Oligomycin A, which is the specific inhibitor of the mitochondrial ATP-hydrolysis that modulates the activity of the c-ring-F0-ATP hydrolase subunit. However, in the case of the SWCNT-COOH, the F0-ATPase binding interaction could be more attenuated by the presence of the carboxyl group.

From the structural point of view, the c-ring-F0-ATP hydrolase subunit represents an uncoupling channel which is part of the mitochondrial permeability transition pore-induced association to mitochondrial dysfunction and apoptosis [15,16]. Following this idea, we suggest that SWCNT could promote the selective inhibition of the F0-ATPase under pathological conditions like cancer where the F0F1-ATPase activity is abnormally exacerbated [16,17,18,19].

In this regard, computational approaches like molecular docking simulation, elastic network models, fractal surface approaches linked to nano-quantitative structure–activity/toxicity relationships (Nano-QSAR/QSTR models), and others [9,20,21,22,23], could be efficiently applied to the exhaustive exploration of the underlying mechanisms of mitochondrial bioenergetic dysfunction (pathological ATP-hydrolysis) from the structural point of view for therapeutic purposes.

Protein structures cannot be investigated using the classical Euclidian mathematical approach. Due to this nature, surface and protein’s chain should be studied using the fractal approach. It is well-known that the fractal dimensions (FDs) are directly associated to the backbone non-Euclidean geometry, as well as to the irregular geometric nature and fractal surface properties of the binding sites (ATPaseF0F1 binding sites). This is explained by the fact that most of the ligand–protein binding processes occur under strict conditions of specificity and, at the same time, that these thermodynamic processes depend on surface phenomena with a defined geometric pattern of stereospecificity and complementarity with the cited binding sites [24]. For this instance, we thus suggest that small changes in the fractal geometry-based surface patterns could directly affect not only the native ATPaseF0F1 binding sites’ folding and solvent accessible surface in the unbound state (unoccupied ATPaseF0F1), but also the conformational entropy and thermodynamic stability of the formed docking complexes generated between the ATPaseF0F1 and the different single-walled carbon nanotubes tested [24,25]. In addition, elastic network models may also be used to study proteins since they may be able to predict global dynamics of proteins and proteins’ complexes [26,27,28]. Thus, these two methods can be used, together with other methods, to study conformational changes induced by SWCNT that may produce harmful effects, inactivation, and so on.

Another relevant approach to study toxicity is computational nano-quantitative structure–activity/toxicity relationships (nano-QSAR/QSTR), which are essential tools to support the discovery process of toxicological effects of nanomaterials (SWCNT). Several approaches have been developed and applied recently to predict potential harmfulness of nanoparticles and nanomaterials [9,10,11,12,13,14,29]. These in silico tools have the quality of being versatile and reconfigurable to many problems. For example, the nano-quantitative structure–binding relationship (Nano-QSBR) models are a type of Nano-QSTR which are able to associate the physico-chemical properties of nanomaterials (nano-descriptors) with the theoretical free energy of binding (FEB values, kcal/mol) obtained from the molecular docking studies and also to experimental nanotoxicological outputs [13,14,30].

Due to this, the QSAR (Nano-QSTR) paradigm has been applied since the beginning of the “nano revolution” as a useful methodology able to support toxicity profiling of nanomaterials and CNT [31,32,33,34,35]. Several approaches by many authors have been reported combining different molecular descriptors, methodologies, and algorithms, including machine learning and deep learning [34,35,36,37,38,39,40,41,42]. In this sense, it is strongly advisable to use Nano-QSTR approaches while performing toxicity profiling of CNT and nanomaterials, since they may be able to predict toxicity as well as directly correlate toxicity/activity with specific features of nanomaterials. In addition, in silico approaches are strongly encouraged by national and supranational authorities in the light of the European Union (EU) 3R principles (replacement, reduction, refinement). Currently, the main limitation of these computational methods is to address a feasible mechanistic interpretation of the nanotoxicity phenomena at the atomic level, in many cases [43]. 

In this work, we propose for the first time a combination of computational modeling approaches, based on molecular docking simulations, elastic network models, fractal surface approaches, and Nano-QSTR calculations, along with experimental validation to tackle the study of binding interactions between single-walled carbon nanotubes with the mitochondrial F0F1-ATPase to contribute to the rational drug design-based nanotechnology, mitotarget drug discovery, and the new area of precision mitochondrial nanomedicine. 

## 2. Materials and Methods

### 2.1. Experimental Section

#### 2.1.1. Reagents and Solutions

Sucrose, ethylene-glycol-bis (b-aminoethyl)-N,N,N0,N0-tetraacetic acid (EGTA), potassium succinate (plus 2 mM rotenone), K2HPO4, and piperazine-N-2-ethanesulfonic acid (Hepes), dimetilsulfóxido (DMSO), and Biuret reagent. All other reagents were commercial products of the highest purity grade available. Single-walled carbon nanotubes like SWCNT-pristine and carboxylated-CNT (SWCNT-COOH) with very low conductivity and semi-metallic properties were provided by Cheaptubes Company (http://cheaptubes.com/shortohcnts.htm) for the execution of experimental in vitro assays using submitochondrial particles. All other reagents were commercial products of the highest purity grade available and were purchased from Sigma-Aldrich products

#### 2.1.2. Carbon Nanotubes’ Characterization

For this instance, a Transmission Electron Microscope (TEM, Tecnai G2-12-SpiritBiotwin FEI-120 kV) was used to characterize the morphology of SWCNT-pristine and oxidized carbon nanotubes such as SWCNT-COOH. The CNT were synthesized by using a catalytic chemical vapor deposition (CCVD) method and functionalized using a concentrated acid mixture of H2SO4:HNO3 mixed (2:1). On the other hand, in order to discover the molecular mechanisms of interaction inhibition of the carbon nanotubes with the F0-ATPase, two types of single-walled carbon nanotubes (SWCNT-pristine and SWCNT-COOH) were modeled by using the Avogadro software, which can be efficiently applied as an advanced molecule editor and visualizer for molecular modeling and computational chemistry. Herein, it is important to note that the in silico analysis was performed just for the purpose of proposing a theoretically rigorous mechanism to explain the potential inhibition of the single-walled carbon nanotubes used on the F0-ATP-ase inhibition. For this reason, the theoretically modeled SWCNTs should not be taken as exact copies from the structural point of view compared with the experimentally tested CNT (SWCNT-pristine and SWCNT-COOH) used in in vitro assays. In this sense, for computational purposes, several approximations were performed mainly based on the diameter and length of carbon nanotubes theoretically modeled compared with those experimentally evaluated, see Figure 1.

#### 2.1.3. Isolation of Rat Liver Submitochondrial Particles (SMP) 

The frozen rat liver mitochondria (RLM) pellet was thawed and diluted with homogenization medium to contain 20 mg of protein/mL. The mitochondrial suspension was subjected to sonic oscillation four times for 15 s with 30 s intervals, using 80 watts at 4 °C [44,45,46,47]. The suspension was then centrifuged at 9750× *g* for 10 min at 4 °C and the submitochondrial particles in the supernatant were isolated by additional centrifugation in a Sorval SV-80 vertical rotor for one hour at 15,000 rev/min at 4 °C, using discontinuous gradient containing 1 mL of 0.5 M sucrose and 1 mL of 2.0 M sucrose in 5 mM Tris-HCl, pH 7.4. Finally, the SMP were suspended in the isolation medium, and the final volume was adjusted to give a stock suspension containing 1 mg of protein/mL.

#### 2.1.4. Standard Incubation Procedure 

Mitochondria liver was isolated and submitochondrial particles (SMP) were energized with 5 mM of potassium succinate (plus 2.5 μM of rotenone) in a standard incubation medium consisting of 125 mM of sucrose, 65 of mM KCl, 2 mM of inorganic phosphate (K_2_HPO_4_), and 10 mM of HEPES-KOH, pH 7.4, at 30 °C [44,45,46,47].

#### 2.1.5. Determination of Mitochondrial F0F1-ATPase Inhibition in Isolated Rat Liver Submitochondrial Particles (SMP)

Isolated rat liver submitochondrial particles (isolated-F0F1-ATPase) (20 mg of protein) were incubated according to the following experimental groups: (1) untreated SMP, (2) SMP + DMSO (100 mM), (3) SMP + SWCNT samples (SWCNT-pristine, SWCNT-COOH) in the range of concentration of 0.5–5 µg/mL, (4) SMP + Oligomycin A (1 µM) as a positive control, and (5) SMP + Oligomycin A (1 µM) + SWCNT samples at 5 µg/mL as an additional control assay. The reactions are started by addition of enzyme, such as H^+^-c-ring/F_0_-ATPase (80 µg of protein). The total volume was 1 mL. After 10 min at 37 °C, the reaction was stopped by addition of 0.5 trichloroacetic acid, 30% (w/v). Phosphate released by ATP hydrolysis is measured on 0.5 mL of molybdate reagent (10 mM ammonium molybdate in 2.5 M sulfuric acid), 1 mL of acetone, and 0.5 mL of 0.4 M citric acid. After each addition, the tubes are homogenized for 10 s in a vortex mixer. The mitochondrial F0F1-ATPase inhibition (F0-ATPase inhibition) for each treatment was calculated by measuring the absorbance at 355 nm [44,45,46,47]. Before all spectrophotometric F0-ATPase inhibition measurements, the blanks with each SWCNT were run and interference absorbance peaks of SWCNT were not observed at 300–400 nm [44,45,46,47]. Furthermore, each SWCNT sample was added under continuous stirring by using magnetic stirrer cuvettes with the aim of preventing the agglomeration process for the SWCNTs during the F0F1-ATPase inhibition assay. For this instance, a tip-sonication regime during 5–10 min was applied which prevents the SWCNT exfoliation into individual SWCNT samples (SWCNT-pristine, SWCNT-COOH), generating a non-agglomerated suspension in monodisperse state before exposure to submitochondrial particle suspension [48,49,50].

#### 2.1.6. Statistical Procedures for the Mitochondrial Assays Using SMP

The one-way analysis of variance (ANOVA) followed by a post hoc Newman–Keuls multiple comparison test was used in order to determine statistical differences between F0-ATPase inhibition assays as independent unrelated experimental groups. In this context, the Newman–Keuls test was used as a multiple and tiered comparison procedure to identify experimental group statistical means that are significantly different from each other from the different experimental conditions evaluated, namely: (i) untreated submitochondrial particles control (SMP as F0-ATPase), (ii) DMSO-treated SMP, (iii) CNT-treated SMP (i.e., SWCNT-pristine or SWCNT-COOH at 1–5 μg/mL), (iv) Oligomycin A-treated SMP (Oligomycin A is a specific F0F1-ATPase inhibitor used as a positive control), and (v) treated SMP mixed with SWCNT or SWCNT-COOH at concentration of 5 μg/mL + Oligomycin A (1 μM). All the biochemical tests, by using isolated rat liver mitochondria (RLM) and submitochondrial particles (SMP), were performed at least three times in triplicate. Normality and variance homogeneity were verified using Shapiro–Wilks and Levene tests respectively, before using one-way ANOVA. In all cases, significance level was set at 5%.

### 2.2. Theoretical Section

#### 2.2.1. Molecular Docking Study 

Docking simulations were performed using Autodock tools mixed Autodock Vina to understand the strength of biochemical interactions across CNT family members (SWCNT-pristine and oxidized-CNT (SWCNT-COOH)) and oligomycin A on F0-ATPase. These in silico binding interactions were performed only to explain hidden biophysical and pharmacodynamic mechanisms observed in the mitochondrial in vitro assays. For this instance, only two types of single walled zigzag SWCNTs (Hamada index *n* = 8, *m* = 0) were modeled, like SWCNT (8.0) and SWCNT-COOH (8.0) as F0F1-ATPase ligands in order to reproduce and model some critical experimental conditions from CNT-properties, like CNT-functionalization linked to observed F0-ATPase inhibition (ATP-hydrolysis inhibition) in isolated RLM and isolated SMP. Following this idea, the F0F1-ATPase C10 ring with oligomycin A from yeast (*Saccharomyces cerevisiae*) as the receptor (protein data bank (PDB) ID: 5BPS, Resolution 2.1Å) was obtained from the *RCSB* Protein Data Bank (PDB) [51]. It is important to note that c-ring-F0-ATPase subunit PDB X-ray structure from *Saccharomyces cerevisiae* (5BPS) can be used in the context of the present docking approaches, taking into account that mitochondrial c-ring-F0-ATPase subunit PDB X-ray structure from *Rattus norvegicus* with oligomycin A has not been crystallized and included in the *RCSB* PDB [51]. However, the oligomycin A pharmacodynamics mechanism is highly conserved in *Rattus norvegicus* according to previous experimental evidences [17].

Before the molecular docking, ATPase C10 ring molecular structure was optimized using the AutoDock Tools 4 software for AutoDock Vina. The algorithm includes the removal of crystallographic water molecules and all the co-crystallized ATPase C10 ring ligand molecules, such as oligomycin A (Oligo A: C_45_H_74_O_11_ ID: EFO) from ATPase C10 ring chains (B, E, K, L, M, O). Oligomycin A is a recognized classical inhibitor of F0F1-ATPase inhibition and it was used as a control to compare the affinity and/or relevant interactions by the re-docking procedure. 

This theoretical algorithm was performed to the c-ring F0-ATPase subunit using a grid box size with dimensions of X = 22 Å, Y = 22 Å, and Z = 22 Å, and the c-ring F0-ATPase subunit grid box center X = 19.917 Å, Y = 19.654 Å, and Z = 29.844 Å to evaluate the interaction of SWCNT + c-ring–F0-ATPase [52], considering the oligomycin A environment to evaluate the SWCNT-surface affinity in the c-ring F0-ATPase subunit active binding site. 

The docking free energy of binding output results (or FEB values) is defined by affinity (like ΔG_bind_ values) for all docked poses of the formed complexes (SWCNT-F0ATPase) and includes the internal steric forces of a given ligand (SWCNT), which can be expressed as the sum of individual molecular mechanics terms of standard chemical potentials as: van der Waals interactions (ΔG_vdW_), hydrogen bond (ΔG_H-bond_), electrostatic interactions (ΔG_electrost_), and intramolecular interactions (ΔG_internal_) ligands (SWCNTs) from empirically validated Autodock Vina scoring function based on default Amber force-field parameters [20,21,22].

#### 2.2.2. Local Perturbation Response Induced by SWCNT on the F0-ATPase Subunit

In parallel with docking simulation, a new elastic network model was performed to propose a potential mechanism based on the SWCNT propensity to perturb the intrinsic motion of F0-ATPase subunit binding residues involved in the docking interactions. For this purpose, the F0-ATPase is represented as a network or graph of the inter-residue contacts from Cα-F0-ATPase atoms of a residue and the overall potential is simply the sum of harmonic potentials between interacting nodes (F0-ATPase residues). The network includes all interactions within a cutoff distance < 4 Å. Information about the orientation of each interaction with respect to the global coordinates system is considered within the force constant matrix and allows prediction of perturbed anisotropic motions [53]. The force constant of the F0-ATPase protein system can be described by a Kirchhoff or Hessian matrix (H_i,j_) to evaluate potential perturbations induced by the SWCNT ligand in the transduction properties of the F0-ATPase enzyme according to the following Equation (1):(1)$$H_{ij}=\left[\begin{array}{c} \begin{array}{ccc} H_{1,1} & H_{1,2}\ldots \ldots \ldots \ldots \ldots \ldots \ldots \ldots \ldots . & H_{1,N}\\ H_{2,1} & H_{2,2}\ldots \ldots \ldots \ldots \ldots \ldots \ldots \ldots \ldots . & H_{2,N}\\ . & .\;\ldots \ldots \ldots \ldots \ldots \ldots \ldots \ldots \ldots \ldots . & . \end{array}\\ \begin{array}{ccc} H_{N,1} & H_{N,2} & \ldots \ldots \ldots \ldots \ldots \ldots \ldots \ldots .\;H_{N.N} \end{array} \end{array}\right]$$
where each *H_i,j_* is a 3 × 3 matrix which holds the anisotropic information regarding the orientation of residues (*i*, *j* nodes). Each such sub-matrix (or the “super element” of the *H_i,j_* Hessian matrix) is defined by the Equation (2) as:(2)$$H_{ij}=\left[\begin{array}{c} \begin{array}{ccc} \delta^{2}V/\delta X_{i}\delta X_{j} & \delta^{2}V/\delta X_{i}\delta Y_{j} & \delta^{2}V/\delta X_{i}\delta Z_{j} \end{array}\\ \begin{array}{ccc} \delta^{2}V/\delta Y_{i}\delta X_{j} & \delta^{2}V/\delta Y_{i}\delta Y_{j} & \delta^{2}V/\delta Y_{i}\delta Z_{j} \end{array}\\ \begin{array}{ccc} \delta^{2}V/\delta Z_{i}\delta X_{j} & \delta^{2}V/\delta Z_{i}\delta Y_{j} & \delta^{2}V/\delta Z_{i}\delta Z_{j} \end{array} \end{array}\right]$$

The second partial derivatives are the harmonic potentials, *V*, between interacting F0-ATPase residues. These partial derivatives are formed by a simple matrix of cosines and the off-diagonal super elements of the *H_i,j_* Hessian matrix are calculated according to Equation (3) as:(3)$$H_{ij}\;=\;\;\;\left[\begin{array}{ccc} -\gamma \frac{\left(X_{j}-X_{i}\right)\left(X_{j}-X_{i}\right)}{S_{i,j}^{2}} & -\gamma \frac{\left(X_{j}-X_{i}\right)\left(y_{j}-Y_{i}\right)}{S_{i,j}^{2}} & -\gamma \frac{\left(X_{j}-X_{i}\right)\left(Z_{j}-Z_{i}\right)}{S_{i,j}^{2}}\\ -\gamma \frac{\left(Y_{j}-Y_{i}\right)\left(X_{j}-X_{i}\right)}{S_{i,j}^{2}} & -\gamma \frac{\left(Y_{j}-Y_{i}\right)\left(Y_{j}-Y_{i}\right)}{S_{i,j}^{2}} & -\gamma \frac{\left(Y_{j}-Y_{i}\right)\left(Z_{j}-Z_{i}\right)}{S_{i,j}^{2}}\\ -\gamma \frac{\left(Z_{j}-Z_{i}\right)\left(X_{j}-X_{i}\right)}{S_{i,j}^{2}} & -\gamma \frac{\left(Z_{j}-Z_{i}\right)\left(Y_{j}-Y_{i}\right)}{S_{i,j}^{2}} & -\gamma \frac{\left(Z_{j}-Z_{i}\right)\left(Z_{j}-Z_{i}\right)}{S_{i,j}^{2}} \end{array}\right]$$
where γ = 0.5 is an interaction constant. The *s_i,j_* is the instantaneous distance between nodes or residues *i* and *j*. The diagonal super elements are calculated by the Equation (4):
(4)$$H_{i,i}=-\overset{N}{\underset{j=1,j\neq i}{\sum }}H_{i,j}$$

Herein, the force constant matrix *H_i,j_* holds information regarding the F0-ATPase-residues position/orientation. The inverse of the Hessian matrix is the covariance matrix of *3N* multi-variant Gaussian distribution, where *p* is an empirical parameter according to the Equation (5) for the new off-diagonal elements of the Hessian matrix which hold the desired information on the residue fluctuations, including the F0-ATPase binding site residues (*i*, *j*) involved in the SWCNT-F0-ATPase docking interactions.
(5)$$H_{ij}\;=\;-\frac{1}{S_{i,j}^{p+2}}\;\left[\begin{array}{c} \begin{array}{c} \left(X_{j}-X_{i}\right)\left(X_{j}-X_{i}\right)\;\left(X_{j}-X_{i}\right)\left(Y_{j}-Y_{i}\right)\;\left(X_{j}-X_{i}\right)\left(Z_{j}-Z_{i}\right) \end{array}\\ \left(Y_{j}-Y_{i}\right)\left(X_{j}-X_{i}\right)\;\left(Y_{j}-Y_{i}\right)\left(Y_{j}-Y_{i}\right)\;\left(Y_{j}-Y_{i}\right)\left(Z_{j}-Z_{i}\right)\\ \left(Z_{j}-Z_{i}\right)\left(X_{j}-X_{i}\right)\;\left(Z_{j}-Z_{i}\right)\left(Y_{j}-Y_{i}\right)\;\left(Z_{j}-Z_{i}\right)\left(Z_{j}-Z_{i}\right) \end{array}\right]$$

Then, we tackle the construction of the local perturbation response scanning maps (LPRS maps) by setting the following conditions: (i) unbound F0-ATPase as the control simulation experiment, (ii) oligomycin A + F0-ATPase, (iii) SWCNT-pristine + F0-ATPase, and (iv) SWCNT-COOH + F0-ATPase.

#### 2.2.3. Performing Nano-QSTR Approaches

The Nano-QSTR models have been developed using a linear regression approach to predict the mitochondrial F0F1-ATPase inhibition values of the SWCNT studied herein. The values used for the development of the continuous model were obtained from molecular docking experiments considering the free energy of binding (FEB values) obtained from the complexes SWCNT-pristine/F0-ATPase and SWCNT-COOH/F0-ATPase. For this purpose, two different sets for both ligands (SWCNT-pristine, SWCNT-COOH) were efficiently built. Considering the three recognized categories of geometric topologies as: zigzag-SWCNT (Hamada index m = 0, n > 0), amchair-SWCNT (Hamada index m = n), and chiral-SWCNT, characterized by the Hamada index (n, m), with m > 0 and m ≠ n, and with its enantiomers (or mirror images), presenting the Hamada index (m, n), which is different from (n, m), with no reflection symmetry [13,14]. Then, regression Nano-QSTR models were developed using the linear regression tool implemented in the Statistica^®^ suite.

The validation of the Nano-QSTR model was performed using the cross-validation module implemented in the software. This procedure is aimed at assessing the predictive accuracy of a model. The test randomly split the dataset into a training set and a validation set, ensuring that if an entry was included in the test set it could not be used in the validation set. In so doing, the model was developed using the cases in the training or learning sample, which, in our study, was 70% of the dataset. The predictive accuracy was then assessed using the remaining 30% of the dataset. In addition, we have also reported the applicability domain (AD) for both models.

Finally, the performance of the model was evaluated using the residuals, *R* and *R^2^*, and other relevant statistics. Regarding the molecular descriptors (MD), we used the DRAGON 7.0^®^ software to calculate the variables that have been used for the development of the models. This software suite is able to calculate up to 7500 different descriptors, belonging to very different classes, such as topological, two-dimensional (2D), three-dimensional (3D), connectivity, and so on [54]. In order to select the best subset of MD, we have performed a feature selection process using a forward stepwise methodology [35] for both models. At the end of this procedure, we were able to develop the pristine and the carboxylate model using respectively two and three MD belonging to the topological class. The two MD used in the SWCNT-pristine model are the Narumi geometric topological index (GNAR) and the electro-topological positive variation (MAXDP). The Narumi index of a graph *G* is defined as the product of the degrees of all its vertices:(6)$$\mathrm{NK}\left(G\right)=\;\overset{n}{\underset{i=1}{\prod }}d_{G}\left(v_{i}\right)$$

The MAXDP is calculated as follows:(7)$$S_{i}=I_{i}+\;∆I_{i=\;}I_{i}+\;\overset{nSK}{\underset{J0i}{\sum }}\frac{I_{i}-I_{j}}{{\left(d_{ij}+1\right)}^{k}}$$
which is calculated as the maximum positive value of Δ*Ii*.

Regarding the SWCNT-COOH model, the continuous model was developed using three MD, one is the same GNAR used for the pristine model. The other two are defined as follows: The first one is the path/walk Randic shape indices that are calculated by summing, over the non-H atoms, the ratios of the atomic path count over the atomic walk count of the same order k and then, dividing by the total number of non-H atoms (nSK). Since path/walk count ratio is independent of molecular size, these descriptors can be considered as measures of molecular shape. Dragon calculates path/walk shape indices from order 2 up to 5, and the index of first order is not provided as the counts of the paths and walks of length one are equal and, therefore, the corresponding molecular index equals one for all molecules. The formula in this case is not reported in the Dragon manual.

Finally, the last molecular descriptor used is the so-called lopping centric index (LOC), which is calculated as the mean information content derived from the pruning partition of a graph: (8)$$\;LOC=\sum k\;\frac{n_{k}}{nSK}*\;log_{2}\frac{n_{k}}{nSK}$$
where *nk* is the number of terminal vertices removed at the *k*th step and *nSK* is the number of non-H atoms.

All the information regarding the descriptors employed in the nano-QSTR models can be retrieved from the Dragon webpage (https://chm.kode-solutions.net/products_dragon_descriptors.php).

## 3. Results and Discussion

### 3.1. CNT Effects on Submitochondrial Particles (SMP)

Herein, we present the in vitro assay on the inhibitory effect of the SWCNT ligands (SWCNT-pristine, SWCNT-COOH) at the range of concentration of 0.5–5 µg/mL over F0-ATPase using isolated rat liver submitochondrial particles (isolated F0F1-ATPase) from mitochondrial inner membrane. In general, we can see that the tested SWCNT exhibit high ability to act as F0-ATPase inhibitors (ATP-hydrolysis) at a range of concentration of 3–5 µg/mL. Besides, a concentration dependence with significant statistical difference (*p* < 0.05) when compared with SMP (untreated SMP group) and the DMSO-treated SMP was observed. We note an oligomycin A-like pattern (positive control group used) for both SWCNT ligands in a range of concentration of 3–5 µg/mL without significant statistical difference (*p* > 0.05) when compared with oligomycin A (Figure 1). According to this, the treated SMP from mixed CNT ligand (5 µg/mL) plus oligomycin A (1 µM) showed the strongest F0-ATPase inhibition (*p* < 0.05) when compared with untreated SMP and the DMSO-treated SMP, and the remaining CNT-treated SMP (3–5 µg/mL). This may suggest a strong synergistic effect on F0-ATPase inhibition (mitochondrial nanotoxicity). Details of these experimental results can be seen in Figure 1.

### 3.2. Modeling F0ATPase Inhibition Induced by SWCNTs

Herein, molecular docking was carried out in order to evaluate the influence of the carbon nanotubes (SWCNT-pristine and SWCNT-COOH) on the F0-ATPase inhibition response. The best docking binding pose from each modeled CNT (SWCNT-pristine, SWCNT-COOH) theoretically suggests that these CNT could act in the same biophysical environment as the oligomycin A based on hydrophobic non-covalent interaction (π-π interactions) involving phenylalanine hydrophobic residues (Phe 55 and Phe 64 of the chains C, D, and M), which are critically involved in the F0-ATPase inhibition (ATP-hydrolysis) in the F0-ATPase subunit active binding site, see Figure 2.

The free energy of binding (FEB) values of the formed docking complexes follow the order: FEB (oligomycin A/F0-ATPase complex) = −9.8 kcal/mol > FEB (SWCNT-COOH/F0-ATPase complex) = −6.8 kcal/mol ~ FEB (SWCNT-pristine complex) = −5.9 kcal/mol, with interatomic distance of interaction lower than 5 Å, in all the cases. Besides, we note the presence of π-π interactions, like Y-shaped and pseudo parallel-displaced motif-orientation preferences, for both single-walled carbon nanotubes. Besides, more electrostatically favored interactions in the CNT-sidewall than the CNT-tips were observed in both simulations (SWCNT-pristine and SWCNT-COOH). This was probably due to better orientation and stability between the planar-benzene-quadrupoles formed between van der Waals surface from the modeled SWCNT and the phenylalanine hydrophobic residues (Phe 55 and Phe 64) of the F0-ATPase binding site and interacting in the same biophysical environment as the F0-ATPase-specific inhibitor (oligomycin A) [17]. Please, see Figure 3.

Next, we carried out the theoretical modeling based on the local perturbation response scanning maps (LPRS maps). The LPRS maps are based on elastic network models (ENM models) and have been widely recognized to study relevant conformational changes promoted from distance-based fluctuations in the alpha carbons (C(α)) of a given target protein (as F0-ATPase under unbound and bound states) at the atomistic and molecular level [53]. It is well-known that the ENM models could explain a large number of the conformational differences based on the perturbation patterns of the network formed by the target residues evaluated (Phe 55 and Phe 64). In this instance, LPRS maps generate comprehensive visualizations of the F0-ATPase inhibition response, which allows to evaluate allosteric signal propagations in response to external perturbations under the presence of a given ligand (i.e., the oligomycin A as a F0-ATPase-specific inhibitor, SWCNT-pristine, and SWCNT-COOH). The results can be seen in Figure 4.

The results on LPRS maps show that both single-walled carbon nanotubes (SWCNT-pristine and SWCNT-COOH) promote a significant change in the perturbation patterns of the network of target residues compared with the physiological condition represented by the unbound state of F0-ATPase. In this regard, we note abrupt perturbations in several blocks of residues more pronounced for the SWCNT-pristine (strong F0-ATPase inhibition) than the SWCNT-COOH (moderate F0-ATPase inhibition) during the interaction with the F0-ATPase. Interestingly, the LPRS map of the SWCNT-pristine/F0-ATPase complex mimicked the toxicodynamic behavior of the oligomycin A/F0-ATPase complex, inducing strong F0-ATPase inhibition (see Figure 4B,C), suggesting a similar pattern of allosteric network perturbation. However, the LPRS map obtained from the SWCNT-COOH/F0-ATPase complex (Figure 4D) exhibits a pattern of perturbation less affected when compared with the physiological condition depicted for the F0F1ATPase unbound state (Figure 4A), maintaining a certain structural and functional coupling between the residues composing the F0-ATPase network, suggesting the presence of a moderate nanotoxicity-based F0-ATPase inhibition. The relevance of these results is that strong local perturbations similar to those observed in Figure 4A, B are able to induce strong allosteric perturbations in the j-effector residues from the F0-ATPase receptor, affecting its mitochondrial catalytic function (ATP-hydrolysis) involving the signal transduction of the perturbations from the block of i-sensor residues which trigger abnormal signals’ propagation across the inter-residue network for j-effector F0-ATPase residues. We could suggest that considering the SWCNT docking position, both ligands (SWCNT-pristine >> SWCNT-COOH) can theoretically disrupt the H^+^-proton flux dynamic in the mitochondrial H^+^-F0-ATPase subunit, compromising the coupling between oxidative phosphorylation and electron transport in the respiratory chain, inducing potential bioenergetic dysfunction and the mitochondria nanotoxicity [9].

In order to quantify potential fractal geometrical perturbations, a fractal surface analysis was carried out to model changes-based perturbations in the geometric surface of the binding effector residues of the F0-ATPase under unbound and bound states (i.e., under SWCNT-pristine and SWCNT-COOH interactions) [9]. Several fractal dimensions (FDs, namely: *D_BW_*, *D_B+BW_*, and *D_W+BW_*) were calculated using the box-counting methods from the LPRS maps previously obtained [55]. The Fractal Theory allows the mathematical modeling of the geometric complexity (across multiple scales) and self-similarity (scale-invariant structure) from non-Euclidean real or virtual objects (such as the tested SWCNT). One of the most important properties in the fractal modeling is the degree of self-similarity. Then, a topological fractal dimension near to 2 is categorized-like, high complexity (i.e., high variety of geometrical information after the docking interaction) and low self-similarity; in contrast, a topological fractal dimension closer to 1 informs about little complexity and high self-similarity after the docking interaction. Herein, the non-Euclidean geometrical patterns were included according to the fractal dimension, like *F**D_BW_*, that describes the surface geometric perturbations in the border of the LPRS map fractal pattern [55]. The *F**D_B+BW_* characterizes the surface geometric perturbations on the white background, and the *F**D_W+BW_* characterizes the fractal perturbations pattern on the black background from the LPRS images calculated for each simulation condition, see Figure 5.

Herein, the obtained FDs are related to the F0-ATPase surface and backbone non-Euclidean geometry [9,55]. FDs inform about how the F0-ATPase folding, packing density, solvent accessibility, and binding interaction properties could be perturbed under the presence of different ligands forming docking complexes (oligomycin A/F0-ATPase complex, SWCNT-pristine/F0-ATPase complex, and SWCNT-COOH/F0-ATPase complex). In this context, we suggest that, in the bound state (i.e., during the docking interaction), the SWCNT-pristine led to higher F0F1-ATPase nanotoxicity-based allosteric perturbations than its carboxylate analogous (SWCNT-COOH) based on their obtained values for the fractal dimensions (*F**D_BW_*), such as SWCNT-pristine/F0-ATPase complex (*F**D_BW_ =* 1.29) < SWCNT-COOH/F0-ATPase complex (*F**D_BW_ =* 1.45), which quantitatively exhibits very similar features-based fractal dimension (*D_BW_*, *D_BBW_*, and *D_WBW_*) compared to physiological condition (unbound F0-ATPase (*F**D_BW_* = 1.45)) used as a control for comparison purposes. It is well-known that slight variations in the fractal dimension as observed in the bound state for the docking complexes SWCNT-pristine/F0-ATPase and SWCNT-COOH/F0-ATPase (Figure 5C, D, respectively) are sufficient to induce changes in the geometry and roughness of the active site of F0-subunit of the F0F1-ATPase. 

These results fit well with the previous LPRS maps, strongly suggesting that the SWCNT-pristine/F0-ATPase complex (*F**D_BW_ =* 1.29) mimicked the nanotoxicological behavior of the specific F0F1-ATPase inhibitor (oligomycin A) with very close calculated fractal dimension for oligomycin A/F0-ATPase complex (*F**D_BW_ =* 1.32), both lower than the physiological condition of unbound F0-ATPase cited above (Figure 5A). As previously cited, a FD ≈ 2 reveals a high variety of geometrical information and low self-similarity, while FD ≈ 1 represents little complexity and high self-similarity. On the other hand, the FD values obtained for *F**D_B+BW_* and *F**D_W+BW_* remain as unperturbed around 1.85 in all the cases, revealing high complexity of geometrical information [9,55].

The results of fractal surface perturbation suggest that the SWCNT-pristine can induce significant changes in the geometrical selectivity of the F0-ATPase, like oligomycin A. It is well-known that perturbation (global and local perturbations) in the three-dimensional spatial arrangement of atoms composing effector residues (*j*-effector allosteric residues) of proteins can be studied using their FDs. Fractal surface perturbations could negatively impact on catalytic function of F0-ATPase, irreversibly affecting the structural properties of the binding cavities, which are of paramount importance in the complementary processes like substrate recognition and ligand geometrical specificity. Probably, topologically perturbed van der Waals fractal surface of F0-ATPase after the docking interaction with SWCNT-COOH could theoretically explain the moderate mitochondrial nanotoxicity observed from the SWCNT-COOH/F0-ATPase docking complex (refer to Figure 4A,D).

Lastly, we carried out a nano-quantitative structure–toxicity relationship approach (Nano-QSRT models) in order to evaluate the influence of additional geometric properties of the ligands SWCNT-pristine and SWCNT-COOH based on the well-known relationship between the topology geometry based on the n, m Hamada index with their nanotoxicological properties (i.e., SWCNT-mitotoxicity).

### 3.3. Performed Nano-QSTR Models

As reported in the Material and Methods Section, the Nano-QSTR model for SWCNT-pristine was developed using only two variables belonging to the topological index category. The observed versus predicted values and the other relevant statistics are reported in the Table 1 and Table 2, and Figure 6, respectively. In addition, we have also reported the AD in Figure 7.

As can be seen in the Table 1 and Table 2, the Nano-QSTR model shows an overall high accuracy and goodness of fit, thus indicating that this model can be used for a continuous prediction of the likelihood of induced mitochondria nanotoxicity inhibition on F0F1-ATPase by interaction with SWCNT-pristine (f(FEB_1)). In this regard, the best Nano-QSTR regression model is based on the linear Equation (9) as:(9)$$f\left(\mathrm{FEB}\_1\right)=-8.24425\left(\mathrm{GNar}\right)+0.614121\left(\mathrm{MAXDP}\right)-2.87142$$

Afterward, we performed a Nano-QSTR model for SWCNT-COOH. For this instance, was carried out a QSTR regression model by using three variables, as in the case of the previous model (i.e., using SWCNT-pristine). Herein, the results obtained on observed versus predicted values, and the other relevant statistical parameters, are summarized in the Table 3 and Table 4, and Figure 8, respectively. In addition, we have also reported the AD in Figure 9.

For the case of the SWCNT-COOH dataset, the final Nano-QSTR regression model to predict the mitochondrial F0-ATPase inhibition (f(FEB_2)) is represented by the linear Equation (10) as:(10)$$f\left(\mathrm{FEB}\_2\right)=-1005.47\left(GNar\right)-1401.69\left(PW5\right)-139.55\left(LOC\right)-2326.4\;\;$$

Overall, the proposed methodologies rigorously obey the Organization for Economic Co-operation and Development (OECD) and the International Organization for Standardization guidelines for development of alternative methods for Computational Nanotoxicology [56].

## 4. Conclusions

In the present study, we presented a combination of experimental and computational approaches to tackle the nanotoxicity of pristine and oxidized single-walled carbon nanotubes (SWCNT-pristine, SWCNT-COOH) based on the mitochondrial F0F1-ATPase inhibition. Experimental evidences supported that the in vitro F0F1-ATPase inhibition responses in submitochondrial particles (SMP) are strongly dependent on the higher level of concentration (from 3 to 5 µg/mL) in both types of single-walled carbon nanotubes (SWCNT-pristine and SWCNT-COOH) evaluated. In addition, both types of carbon nanotubes show an interaction inhibition pattern for the F0F1-ATPase enzyme, similar to the oligomycin A (specific F0F1-ATPase inhibitor). On the other hand, the best binding pose for the obtained complexes fit well with the previous experimental results. The free energy of binding (FEB values) for the formed docking complexes followed the affinity order: FEB (oligomycin A/F0-ATPase complex) = −9.8 kcal/mol > FEB (SWCNT-COOH/F0-ATPase complex) = −6.8 kcal/mol ~ FEB (SWCNT-pristine complex) = −5.9 kcal/mol, with relevant interatomic distance of interaction lower than 5 Å, in all the cases, and with predominance of van der Waals hydrophobic interactions with critical F0-ATPase binding site residues (Phe 55 and Phe 64) belonging to the same biophysical environment as the oligomycin A inhibitor. In addition, results on elastic network models (LPRS maps) show that the SWCNT-pristine can promote an abrupt perturbation in several blocks of residues (strong F0-ATPase nanotoxicity inhibition), more pronounced than the analogous SWCNT-COOH (moderate F0-ATPase nanotoxicity inhibition), triggering perturbations on the allosteric responses and abnormal signals’ propagation across the inter-residue network of the F0F1-ATPase. In accordance with this, results on the fractal surface of interactions based on the formed docking complexes (SWCNT-pristine/F0F1-ATPase >> SWCNT-COOH/F0F1-ATPase) suggest that the SWCNT-pristine interactions topologically affect the van der Waals fractal surface and geometric properties of F0-ATPase compared to physiological condition (unbound F0-ATPase). We suggest that the SWCNT-pristine perturbations could negatively impact on catalytic function of F0-ATPase (mitochondrial ATP-hydrolysis), by irreversibly affecting the structural properties of the binding cavities in the F0-subunit. Lastly, the predictive Nano-QSTR models showed that a linear correlation between SWCNT topology and the nanotoxicity induced was present and can be predicted using a Nano-QSTR approach.

Finally, these results open new opportunities toward to the better understanding of the molecular nanotoxicity mechanisms, relevance of mitotarget drug discovery, and rational drug design-based nanotechnology with potential biomedical application in precision nanomedicine.

## Figures and Tables

**Figure 1 biology-10-00171-f001:**
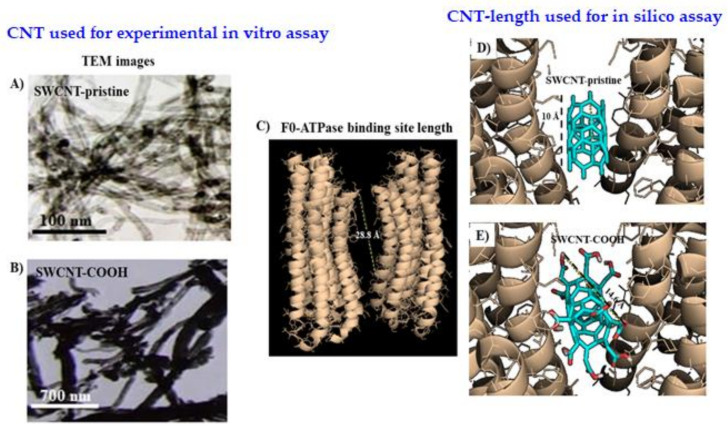
On the right, Transmission Electron Microscope (TEM) images obtained of carbon nanotubes, such as (**A**) SWCNT-pristine and (**B**) SWCNT-COOH used in this study for the experimental in vitro assay. On the left, (**C**) representation of the length of the unoccupied F0-ATPase binding site, (**D**,**E**) representation of the lengths of the theoretically modeled SWCNT-pristine and SWCNT-COOH within the F0-ATPase used for the in silico assay of F0-ATPase inhibition. Additional details can be found in the Appendix A.

**Figure 2 biology-10-00171-f002:**
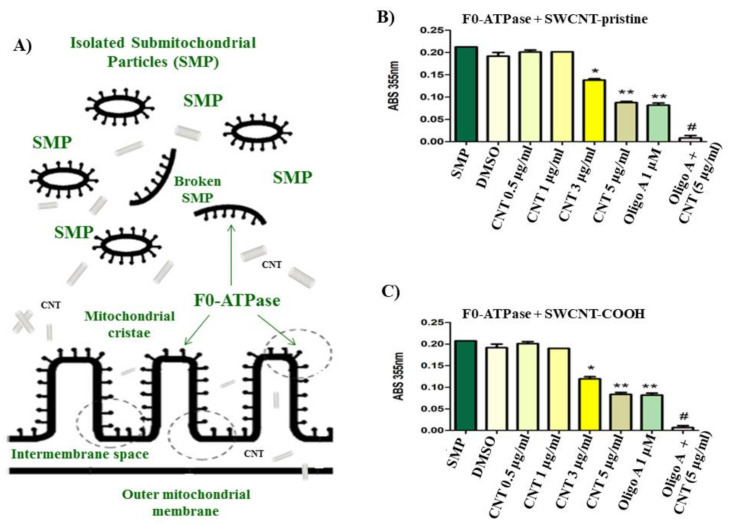
(**A**) Schematic representation of carbon nanotubes (CNT) interacting with isolated rat liver submitochondrial particles (SMP as F0-ATPase). (**B**,**C**) Results of experimental in vitro evaluation of the F0-ATPase inhibition induced by CNT (i.e., CNT as SWCNT-pristine or SWCNT-COOH) using F0-ATPase under the different conditions described in the Material and Methods Section as treatments: (i) untreated submitochondrial particles control (SMP as F0-ATPase), (ii) DMSO-treated SMP, (iii) CNT-treated SMP (1–5 µg/mL), (iv) oligomycin A-treated SMP (oligomycin A is a specific F0F1-ATPase inhibitor used as a positive control), and (v) treated SMP mixed with SWCNT or SWCNT-COOH at concentration of 5 µg/mL + oligomycin A (1 µM) to mimick synergistic effects on F0-ATPase inhibition, which was performed as an additional control group. Results are representative of three experiments (n = 3). Symbols (*, **, #) were used to denote statistical differences (*p* < 0.05) between the evaluated experimental groups used in the in vitro assay containing the SMP.

**Figure 3 biology-10-00171-f003:**
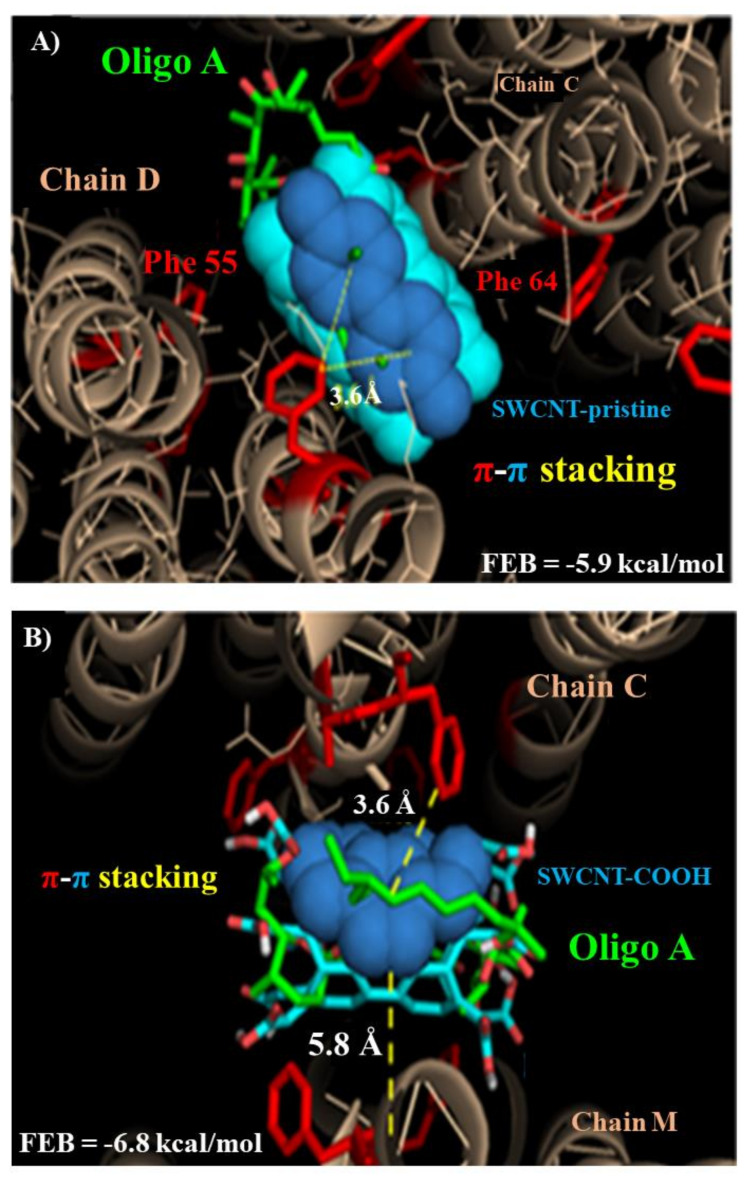
Snapshots selection from molecular docking interactions obtained from the best binding poses of the ligands as (**A**) superimposed representation of oligomycin A and SWCNT-pristine, and (**B**) superimposed representation of oligomycin and SWCNT-pristine and SWCNT-COOH interacting with critical phenylalanine hydrophobic residues (Phe 55 and Phe 64: labelled red) which belong to the target chains C, D, and M in the F0-ATPase subunit receptor. Please note that oligomycin A (labelled green) corresponds to the control simulation experiment used here as a reference due to this ligand being the specific inhibitor of the F0-ATPase in all cases.

**Figure 4 biology-10-00171-f004:**
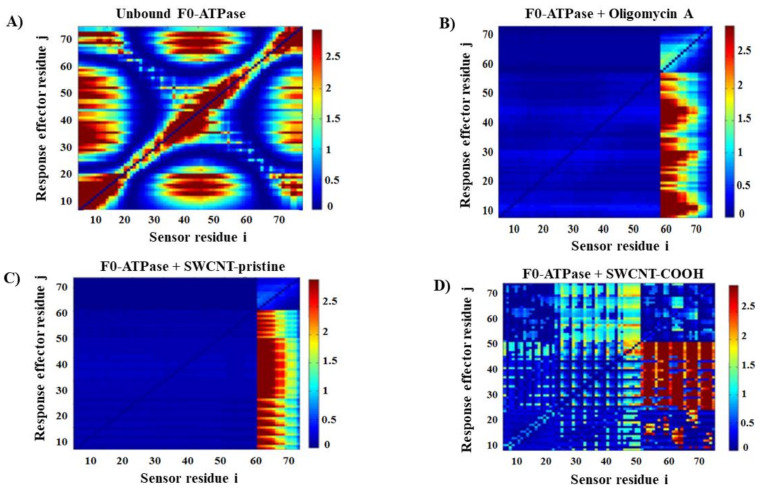
Perturbation response analysis for the F0-ATPase inhibition response. (**A**) LPRS map obtained for the unbound F0-ATPase as the control simulation experiment. Individual LPRS maps obtained from the best docking complexes (in the bound state for all the ligands tested) with intensity bar color representing the i,j residue perturbations (on the right) for: (**B**) oligomycin A/F0-ATPase complex, (**C**) SWCNT-pristine/F0-ATPase complex, and (**D**) SWCNT-COOH/F0-ATPase complex. All the LPRS maps were established in range of the low-frequency normal modes in order to capture relevant fluctuations associated with F0-ATPase catalytic function across the different conditions simulated.

**Figure 5 biology-10-00171-f005:**
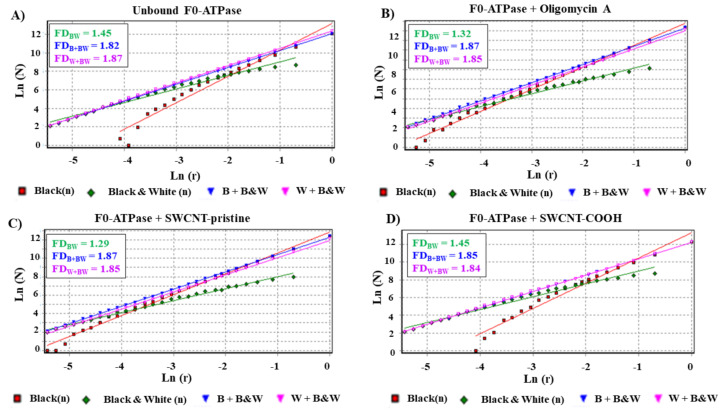
Fractal spectrum based on the box-counting method performed to obtain the slopes of the linear regression yields from binary black/white LPRS maps image-processing. These slopes represent the fractal dimensions (FD: *D_BW_*, *D_B+BW_*, and *D_W+BW_*) for the best docking complexes, namely: (**A**) unbound F0-ATPase, (**B**) oligomycin A/F0-ATPase complex, (**C**) SWCNT-pristine/F0-ATPase complex, and (**D**) SWCNT-COOH/ F0-ATPase complex.

**Figure 6 biology-10-00171-f006:**
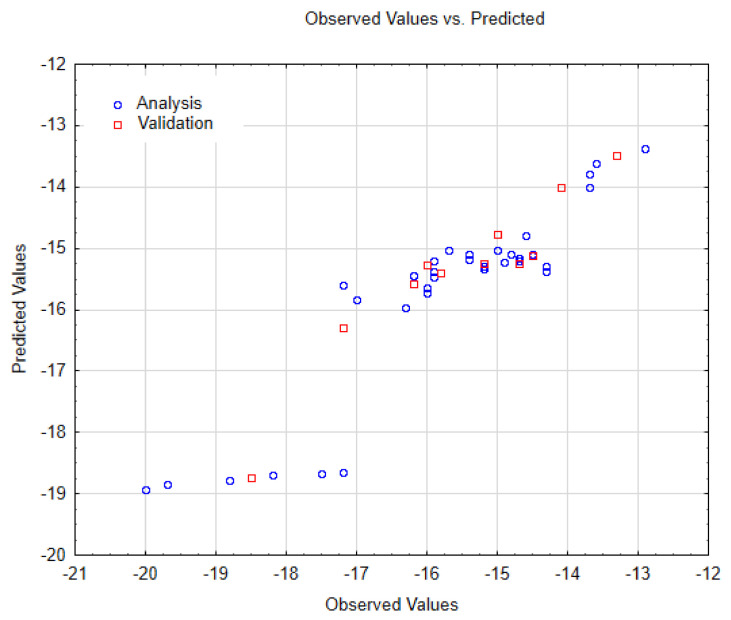
Results of observed versus predicted values obtained for the Nano-QSTR regression model performed for the SWCNT-pristine data.

**Figure 7 biology-10-00171-f007:**
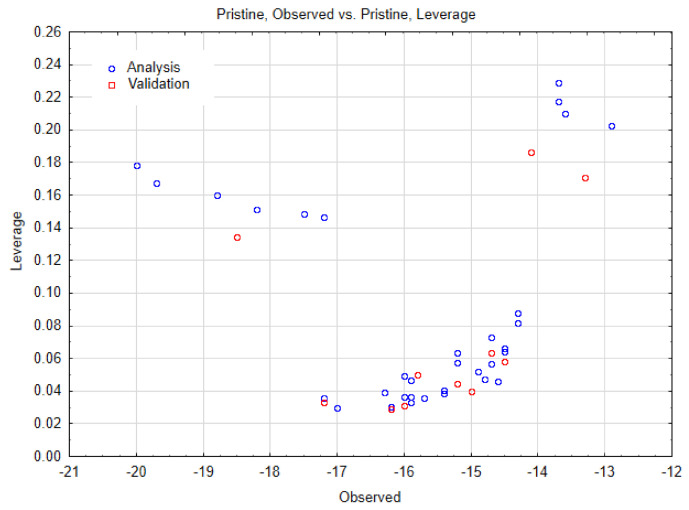
Applicability domain for SWCNT-pristine data.

**Figure 8 biology-10-00171-f008:**
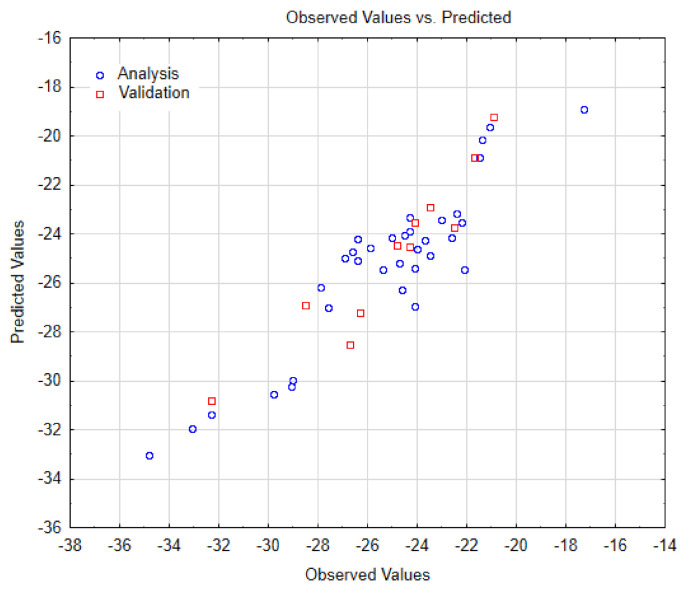
Results of observed versus predicted values obtained for the Nano-QSTR regression model performed for the SWCNT-COOH data.

**Figure 9 biology-10-00171-f009:**
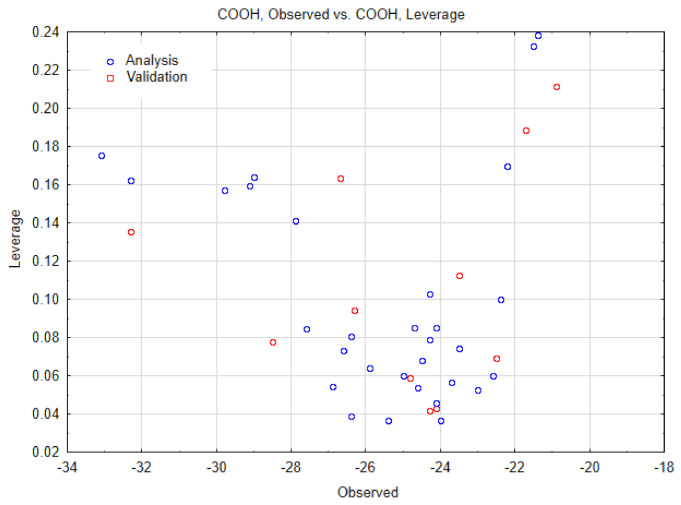
Applicability domain for SWCNT-COOH data.

**Table 1 biology-10-00171-t001:** Results of the Nano-QSTR regression model for mitochondrial F0-ATPase inhibition induced by SWCNT-pristine.

SWCNT-Pristine (n, m)	Data Observed	Data Predicted	Data Residuals	Cross-Validation ^(a,b)^
amchair 3.3	−20.00000	−18.93350	−1.06650	training
amchair 4.4	−19.70000	−18.83954	−0.86046	training
amchair 5.5	−18.80000	−18.77444	−0.02556	training
amchair 6.6	−18.50000	−18.72592	0.22592	validation
amchair 7.7	−18.20000	−18.68908	0.48908	training
amchair 8.8	−17.50000	−18.66083	1.16083	training
amchair 9.9	−17.20000	−18.63872	1.43872	training
chiral 3.2	−17.20000	−16.28865	−0.91135	validation
chiral 4.1	−17.20000	−15.58908	−1.61092	training
chiral 4.2	−17.00000	−15.84427	−1.15573	training
chiral 4.3	−16.30000	−15.96788	−0.33212	training
chiral 5.1	−16.20000	−15.56891	−0.63109	validation
chiral 5.2	−16.20000	−15.44925	−0.75075	training
chiral 5.3	−16.00000	−15.63349	−0.36651	training
chiral 5.4	−16.00000	−15.72809	−0.27191	training
chiral 6.1	−16.00000	−15.26864	−0.73136	validation
chiral 6.2	−15.90000	−15.19863	−0.70137	training
chiral 6.3	−15.90000	−15.37446	−0.52554	training
chiral 6.4	−15.90000	−15.47126	−0.42874	training
chiral 6.5	−15.80000	−15.39773	−0.40227	validation
chiral 7.1	−15.70000	−15.02346	−0.67654	training
chiral 7.2	−15.40000	−15.17288	−0.22712	training
chiral 7.3	−15.40000	−15.10112	−0.29888	training
chiral 7.4	−15.20000	−15.24345	0.04345	validation
chiral 7.5	−15.20000	−15.34078	0.14078	training
chiral 7.6	−15.20000	−15.28875	0.08875	training
chiral 8.1	−15.00000	−15.03210	0.03210	training
chiral 8.2	−15.00000	−14.77537	−0.22463	validation
chiral 8.3	−14.90000	−15.23422	0.33422	training
chiral 8.4	−14.80000	−15.08583	0.28583	training
chiral 8.5	−14.70000	−15.15833	0.45833	training
chiral 8.6	−14.70000	−15.24567	0.54567	validation
chiral 8.7	−14.70000	−15.20136	0.50136	training
chiral 9.3	−14.60000	−14.79209	0.19209	training
chiral 9.4	−14.50000	−15.12489	0.62489	training
chiral 9.5	−14.50000	−15.11612	0.61612	validation
chiral 9.6	−14.50000	−15.09848	0.59848	training
chiral 9.7	−14.30000	−15.29203	0.99203	training
zigzag 3.0	−14.30000	−15.37860	1.07860	training
zigzag 4.0	−14.10000	−14.00654	−0.09346	validation
zigzag 5.0	−13.70000	−14.00654	0.30654	training
zigzag 6.0	−13.70000	−13.79221	0.09221	training
zigzag 7.0	−13.60000	−13.62087	0.02087	training
zigzag 8.0	−13.30000	−13.48024	0.18024	validation
zigzag 9.0	−12.90000	−13.36294	0.46294	training

Datasets: training ^(a)^ and validation sets ^(b)^.

**Table 2 biology-10-00171-t002:** Results of the relevant statistical parameters obtained from the Nano-QSTR regression model for SWCNT-pristine.

Statistical Parameters	Value
Multiple R	0.911445
Multiple R²	0.830731
Adjusted R²	0.819811
Sum of squares Model	77.70196
Degrees of freedom Model	2
Mean squared errors Model	38.85098
Sum of squares Residual	15.83245
Degrees of freedom Residual	31
Mean squared errors Residual	0.510724
F	76.07035
P	0

**Table 3 biology-10-00171-t003:** Results of the Nano-QSTR regression model for mitochondrial F0-ATPase inhibition induced by SWCNT-COOH.

SWCNT-COOH (n, m)	Data Observed	Data Predicted	Data Residuals	Cross-Validation ^(a,b)^
amchair 3.3	−34.80000	−33.04305	−1.75695	training
amchair 4.4	−33.10000	−31.92664	−1.17336	training
amchair 5.5	−32.30000	−31.36844	−0.93156	training
amchair 6.6	−32.30000	−30.81023	−1.48977	validation
amchair 7.7	−29.80000	−30.53113	0.73113	training
amchair 8.8	−29.10000	−30.25203	1.15203	training
amchair 9.9	−29.00000	−29.97293	0.97293	training
chiral 3.2	−28.50000	−26.92826	−1.57174	validation
chiral 4.1	−27.90000	−26.16358	−1.73642	training
chiral 4.2	−27.60000	−27.01059	−0.58941	training
chiral 4.3	−26.90000	−25.00316	−1.89684	training
chiral 5.1	−26.70000	−28.51704	1.81704	validation
chiral 5.2	−26.60000	−24.71436	−1.88564	training
chiral 5.3	−26.40000	−25.08549	−1.31451	training
chiral 5.4	−26.40000	−24.19447	−2.20553	training
chiral 6.1	−26.30000	−27.20385	0.90385	validation
chiral 6.2	−25.90000	−24.54620	−1.35380	training
chiral 6.3	−25.40000	−25.47553	0.07553	training
chiral 6.4	−25.00000	−24.13724	−0.86276	training
chiral 6.5	−24.80000	−24.44496	−0.35504	validation
chiral 7.1	−24.70000	−25.21534	0.51534	training
chiral 7.2	−24.60000	−26.28422	1.68422	training
chiral 7.3	−24.50000	−24.04522	−0.45478	training
chiral 7.4	−24.30000	−24.52728	0.22728	validation
chiral 7.5	−24.30000	−23.88676	−0.41324	training
chiral 7.6	−24.30000	−23.32855	−0.97145	training
chiral 8.1	−24.10000	−26.95336	2.85336	training
chiral 8.2	−24.10000	−23.51211	−0.58789	validation
chiral 8.3	−24.10000	−25.39320	1.29320	training
chiral 8.4	−24.00000	−24.63822	0.63822	training
chiral 8.5	−23.70000	−24.27680	0.57680	training
chiral 8.6	−23.50000	−22.90990	−0.59010	validation
chiral 8.7	−23.50000	−24.89841	1.39841	training
chiral 9.3	−23.00000	−23.42979	0.42979	training
chiral 9.4	−22.60000	−24.13724	1.53724	training
chiral 9.5	−22.50000	−23.71859	1.21859	validation
chiral 9.6	−22.40000	−23.16039	0.76039	training
chiral 9.7	−22.20000	−23.53151	1.33151	training
zigzag 3.0	−22.10000	−25.42752	3.32752	training
zigzag 4.0	−21.70000	−20.86901	−0.83099	validation
zigzag 5.0	−21.50000	−20.86901	−0.63099	training
zigzag 6.0	−21.40000	−20.17126	−1.22874	training
zigzag 7.0	−21.10000	−19.61305	−1.48695	training
zigzag 8.0	−20.90000	−19.19440	−1.70560	validation
zigzag 9.0	−17.30000	−18.91530	1.61530	training

Datasets: training ^(a)^ and validation sets ^(b)^.

**Table 4 biology-10-00171-t004:** Results of the relevant statistical parameters obtained from the Nano-QSTR regression model for SWCNT-COOH.

Statistical Parameters	Value
Multiple R	0.918915
Multiple R²	0.844404
Adjusted R²	0.828845
Sum of squares Model	366.1187
Degrees of freedom Model	3
Mean squared errors Model	122.0396
Sum of squares Residual	67.46364
Degrees of freedom Residual	30
Mean squared errors Residual	2.248788
F	54.26905
*p*	0.000000

## Data Availability

The data that support the findings of this study are available from the corresponding author, upon reasonable request.

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
