# Peer review of "New Mechanistic Insights on Carbon Nanotubes’ Nanotoxicity Using Isolated Submitochondrial Particles, Molecular Docking, and Nano-QSTR Approaches"

_biology, 2021, doi:10.3390/biology10030171_

Round 1
Reviewer 1 Report
The authors report an interesting work on carbon nanotubes nanotoxicity using a combination of experimental and computational studies. The reported results are suitable for the audience of this journal and can be considered for publication after moderate revision. In some cases the equations and figures appear to be shifted or stretched, this need correction for a better presentation of the results. In Fig.3 the x and y axis can not be visualized well and need to be adjusted. A similar comment for Fig.4. Similarly for tables 1-4, these seem print-screen and need to be converted in a proper table format. In Fig.5-6, can the authors include the error bar parameter? The captions of Fig.5 and Fig.6 need to be improved to provide more information to the reader. The quality of Fig.1 can be also improved to allow a better and prompt visualization of the results. Have the authors verified the role of carbon nanotubes-length in the presented results?
Author Response
Editor May Tang,Assistant Editor,MDPI Biology Editorial Office, Dear Dr.
Please, find attached a revised version of the manuscript entitled: “New Mechanistic Insights on Carbon Nanotubes Nanotoxicity Using Isolated Submitochondrial Particles, Molecular Docking, and Nano-QSTR approaches.”, by the authors: Michael González-Durruthy1,2,*, Riccardo Concu1,*, Juan M. Ruso 2, M. Natália D.S. Cordeiro1 (Manuscript ID: Biology-1077594).
We wish to acknowledge the reviewer’s comments and the expediency of the review process. Below, we provide a detailed response to the individual comments by the reviewers and list all changes implemented in the revised manuscript.
Finally, we hope that the revised manuscript can now be considered suitable for publication in Biology. Looking forward to hearing from you.Yours sincerely,Michael González-Durruthy
Riccardo Concu
Reviewer 1:
The authors report an interesting work on carbon nanotubes nanotoxicity using a combination of experimental and computational studies. The reported results are suitable for the audience of this journal and can be considered for publication after moderate revision. In some cases, the equations and figures appear to be shifted or stretched, this need correction for a better presentation of the results. In Fig.3 the x and y axis can not be visualized well and need to be adjusted. A similar comment for Fig.4. Similarly, for tables 1-4, these seem print-screen and need to be converted in a proper table format. In Fig.5-6, can the authors include the error bar parameter? The captions of Fig.5 and Fig.6 need to be improved to provide more information to the reader. The quality of Fig.1 can be also improved to allow a better and prompt visualization of the results. Have the authors verified the role of carbon nanotubes-length in the presented results?
Question 1: In some cases, the equations and figures appear to be shifted or stretched, this need correction for a better presentation of the results. In Fig.3 the x and y axis can not be visualized well and need to be adjusted. A similar comment for Fig.4. Similarly, for tables 1-4, these seem print-screen and need to be converted in a proper table format. In Fig.5-6, can the authors include the error bar parameter? The captions of Fig.5 and Fig.6 need to be improved to provide more information to the reader. The quality of Fig.1 can be also improved to allow a better and prompt visualization of the results. Have the authors verified the role of carbon nanotubes-length in the presented results?
Answer 1: We follow all the reviewer's suggestions and the equations, figures, and tables were corrected in the revised version of the manuscript to provide clarity and best understanding for the readers. Please, also note that was necessary to include an additional Figure 1 to clarify the question about the carbon nanotube-lengths in the context of the in vitro and in silico experiments of interactions with the F0-ATPase. Then, following this new information the all the Figure were re-numbered in the revised version of the manuscript from Figure 1 to Figure 7.
Reviewer 2
Dear authors, this is very important and interesting topic. The first sentence in Abstract maked me interesting, the second – it seems to be very good work. However, next paragraphs & Figures, sorry, but how can you underline the SWCNT while your experimental material are very large multiwalled CNTs (Fig.1)? Unfortunately, there is no word about the source of the tubes.
The scale problem: SWNCT are at least 100x tinier then those presented if Fig.1. On the other hand are much longer than those presented in scheme in Figs 1&2.
On the other hand authors describe computational details, why?
These are implemented in the software. Finally biological problem: how to force CNT to be localized exactly in these cell’s regions. Moreover, the values of interaction energies (I) small – CNT, (II) kcal/mol per “mol” of what?
Thus, the work is unacceptable.
Question 1: Dear authors, this is very important and interesting topic. The first sentence in Abstract maked me interesting, the second – it seems to be very good work. However, next paragraphs & Figures, sorry, but how can you underline the SWCNT while your experimental material are very large multiwalled CNTs (Fig.1)? Unfortunately, there is no word about the source of the tubes. The scale problem: SWNCT are at least 100x tinier then those presented if Fig.1. On the other hand are much longer than those presented in scheme in Figs 1&2
Answer 1: We appreciate very much the reviewer's comments, acknowledging that this is a topic of great relevance to ensure the safe use of carbon nanomaterials (single-walled carbon nanotubes as SWCNT-pristine and SWCNT-COOH) with potential biomedical applications. Regarding the objections raised by the reviewer, let us clarify these issues one by one, due to we note that the reviewer was unable to understand the sense and proposed methodologies of the current study minly concerning the in silico approaches proposed to support our hypothesis on the inhibition may be induced by the SWCNT-pristine and SWCNT-COOH on the mitocondrial F0-ATPase enzyme.
Firstly, it is important to clarify that the carbon nanotubes used were single-walled carbon nanotubes (SWCNT-pristine and SWCNT-COOH) and not multi-walled as stated by the reviewer. In this regard, to clarify this point a new paragraph and Figure 1 were added in the revised version of the manuscript marked in green in the Material and Methods section as:
Carbon nanotubes characterization
Single-walled carbon nanotubes like SWCNT-pristine and carboxylated-CNT (SWCNT-COOH) with very low conductivity and semi-metallic properties were provided by Cheaptubes Company (http://cheaptubes.com/shortohcnts.htm) for the execution of experimental in vitro assays using submitochondrial particles.
For this instance, a Transmission Electron Microscope (TEM, Tecnai G2-12 - SpiritBiotwin FEI - 120 kV) was used to characterize the morphology of SWCNT-pristine and oxidized carbon nanotubes as SWCNT-COOH. The CNT were synthesized by using a CCVD method and functionalized using a concentrated acid mixture of H2SO4:HNO3 mixed (2:1). By the other hand, in order to deep into the molecular mechanisms of interaction inhibition of the carbon nanotubes with the F0-ATPase, two types of single walled carbon nanotubes (SWCNT-pristine and SWCNT-COOH) were modeled by using the Avogadro software which can be efficiently applied as an advanced molecule editor and visualizer for molecular modeling and computational chemistry. Herein, it is important to note that the in silico analysis was performed just for the purpose of propose a theoretically rigorous mechanism to explain the potential inhibition of the single-walled carbon nanotubes used on the F0-ATP-ase inhibition. By this reason the theoretically modeled SWCNTs should not be taken as exact copies from the structural point of view compared with the experimentally tested CNT (SWCNT-pristine and SWCNT-COOH) used in vitro assays. In this sense, computational purposes several approximations were performed mainly based on the diameter and length of carbon nanotubes theoretically modeled compared with the experimentally evaluated. See Figure 1.
Figure 1. On the right, TEM images obtained of carbon nanotubes as A) SWCNT-pristine, and B) SWCNT-COOH used in this study for the carried out the experimental in vitro assay. On the left, C) Representation of the length of the unoccupied F0-ATPase binding site, D) and E) representation of the lengths of the theoretically modeled SWCNT-pristine and SWCNT-COOH within the F0-ATPase used for the in silico assay of F0-ATPase inhibition.
Besides, we must emphasize that in the case of experimental in vitro assays the single-walled carbon nanotubes were incorporated forming a monodisperse system before exposure to submitochondria particle suspension by using continuous stirring with magnetic stirrer cuvettes and a tip-sonication regime during 5-10 min to avoid the spontaneous agglomeration phenomena before the in vitro exposure of the SMP (i.e., mitochondrial F0-ATPase enzyme). Then, it is important to clarify that the TEM-image showed in the study correspond to the isolated samples of the SWCNT-pristine and SWCNT-COOH in the total absence of SMP, and in this sense it is just referred to the isolated carbon nanotube characterization without SMP.
Question 2: On the other hand, authors describe computational details, why? These are implemented in the software.
Answer 2: We appreciate very much the reviewer's comments about the computational details. However, it is important to highlight that theoretical details were described in order to confer biophysical and biochemical meaning to multiple variables (physic-chemical nanodescriptors like: diameter, size, charge, n,m-Hamada indices, electro-topological properties, etc.) involved in the interactions of the single-walled carbon nanotubes and the network of residues forming the F0-ATPase enzyme which are involved in the docking interactions following non-linear responses. Then, we consider that these computational details cannot be ignored and must be explicitly addressed in context of the in silico study of the toxicodynamic behavior of nanoparticles (SWCNT-pristine and SWCNT-COOH) which differ significantly of the traditional xenobiotics (conventional drugs) from the structural point of view and where computational methodologies (i.e., molecular docking, QSAR, etc.) can be used without any modification or adaptation to study potential molecular interactions.
Question 3: Finally, biological problem: how to force CNT to be localized exactly in these cell’s regions.
Answer 3: The reviewer raise an interesting point concerning the carbon nanotubes specific interactions. In this regard, please note that the experimental part of the study was addressed by using isolated submitochondrial particles (F0-ATPase enzyme). Then for this instance we don't use the whole cell. Besides, we use only the inner mitochondrial membrane where are embedded the F0-ATPase enzymes (refer to Figure 2) which can be isolated as submitochondrial particles maintaining its catalytic activity for 3-5 hours at 4°C. Herein, it is important to highlight that the SMP have been widely used and recognized to the exploration of molecular mechanisms of mitotoxicity during pharmacology and toxicology in vitro assays. Then, the proposed protocol unequivocally ensure that the interaction of the SWCNT-pristine and SWCNT-COOH will happen specifically and directly with the F0-ATPase target enzymes. Additionally, the reviewer can verify that the experimental assays with SMP (F0F1-ATPase) were rigorously conducted in the presence of the classical or specific inhibitor of F0-ATPase (Oligomycin A) as a reference control group in order to express the maximum inhibitory response of F0F1-ATP-ase, and the results clearly follow a dose-dependent pattern from 0.5-5 µg / ml concentrations for both single-walled carbon nanotubes used when compared with the untreated-submitochondrial particles (SMP as F0-ATPase) and DMSO-treated SMP control groups. Furthermore, the F0-ATP inhibition induced by the nanotubes was also corroborated in the additional control group formed by the treated-SMP+ Oligomycin A(1µM) + mixed with SWCNT or SWCNT-COOH at concentration 5µg/ml to mimicking a synergistic effect on F0-ATPase inhibition represented by the maximum decrease in absorbance (Abs) from 355 nm compared with the untreated-submitochondrial particles (SMP as F0-ATPase) and DMSO-treated SMP control groups.
Question 4: Moreover, the values of interaction energies (I) small – CNT, (II) kcal/mol per “mol” of what?
Answer 4: Herein, it is not clear what exactly the reviewer asks. So we assume that the reviewer try to ask for units of the theoretical free energy of binding (FEB) which are used during the docking experiments to quantify the thermodynamic interaction between ligand-proteins in terms of “kcal/mol”. In this context, ΔGbind < 0 kcal/mol denote the presence of spontaneous thermodynamic process, otherwise denotes partial or total absence of docking interaction without docking complex formation between the ligands and protein.
Reviewer 3
Question 1: The paper is so verbose that it becomes impossible to follow the discussion of the sole paper. It contains so much general information that could be moved to the supplementary or summarized without the loss of the integrity of the paper. Authors are encouraged to reduce the size of their paper to a maximum of 12-15 pages and move all general information to the SI.
Answer 1: We follow all the reviewer's suggestions and we removed irrelevant information (equations 1 and 2) mainly in the Material and methods in the subsection Molecular docking study in the revised version of the manuscript.
Question 2: The processing of SWCNT is not clearly demonstrated. As the authors correctly mentioned the SWCNT tends to agglomerate owing to the high van-der-Waals interactions and without good dispersion, the discussion on the interaction of SWCNTs with other species is erroneous. TEM images on page 9 of the paper clearly demonstrated the heavily agglomerated area of SWCNTs which adversely affected their interaction with other moieties.
Question 3: It is not cleared if the authors do the experimental works by themselves or they just model some other work data. If they perform their own experiments, they should state the chemical and their grade which are using.
Question 5: The authors used SWNT-COOH for their modeling and it is not cleared how to interpret this structure. Normally oxidation in SWCNTs induced by acid-treatment and the outer surface of SWCNT will be covalently covered with COOH. In practice, the smaller SWCNTs oxidized more (contain more -COOH) compared to the larger ones. It is also not clear who the aspect ratio of SWCNTs affected their interactions.
Answer 2, 3 and 5: We agree with the reviewer's 3 observation. Please, refer to the same explanation provided for the reviewer 2 as:
Carbon nanotubes characterization.
Single-walled carbon nanotubes like SWCNT-pristine and carboxylated-CNT (SWCNT-COOH) with very low conductivity and semi-metallic properties were provided by Cheaptubes Company (http://cheaptubes.com/shortohcnts.htm) for the execution of experimental in vitro assays using submitochondrial particles.
For this instance, a Transmission Electron Microscope (TEM, Tecnai G2-12 - SpiritBiotwin FEI - 120 kV) was used to characterize the morphology of SWCNT-pristine and oxidized carbon nanotubes as SWCNT-COOH. The CNT were synthesized by using a CCVD method and functionalized using a concentrated acid mixture of H2SO4:HNO3 mixed (2:1). By the other hand, in order to deep into the molecular mechanisms of interaction inhibition of the carbon nanotubes with the F0-ATPase, two types of single walled carbon nanotubes (SWCNT-pristine and SWCNT-COOH) were modeled by using the Avogadro software which can be efficiently applied as an advanced molecule editor and visualizer for molecular modeling and computational chemistry. Herein, it is important to note that the in silico analysis was performed just for the purpose of propose a theoretically rigorous mechanism to explain the potential inhibition of the single-walled carbon nanotubes used on the F0-ATP-ase inhibition. By this reason the theoretically modeled SWCNTs should not be taken as exact copies from the structural point of view compared with the experimentally tested CNT (SWCNT-pristine and SWCNT-COOH) used in vitro assays. In this sense, for computational purposes several approximations were performed mainly based on the diameter and length of carbon nanotubes theoretically modeled compared with the experimentally evaluated. See Figure 1.
Figure 1. On the right, TEM images obtained of carbon nanotubes as A) SWCNT-pristine, and B) SWCNT-COOH used in this study for the carried out the experimental in vitro assay. On the left, C) Representation of the length of the unoccupied F0-ATPase binding site, D) and E) representation of the lengths of the theoretically modeled SWCNT-pristine and SWCNT-COOH within the F0-ATPase used for the in silico assay of F0-ATPase inhibition.
Besides, we must emphasize that in the case of experimental in vitro assays the single-walled carbon nanotubes were incorporated forming a monodisperse system before exposure to submitochondria particle suspension by using continuous stirring with magnetic stirrer cuvettes and a tip-sonication regime during 5-10 min to avoid the spontaneous agglomeration phenomena before the in vitro exposure of the SMP (i.e., mitochondrial F0-ATPase enzyme). Then, it is important to clarify that the TEM-image showed in the study correspond to the isolated samples of the SWCNT-pristine and SWCNT-COOH in the total absence of SMP, and in this sense it is just referred to the isolated carbon nanotube characterization without SMP.
Authors should apply more robust algorithms like k-mean clustering for the classification of their data.
Answer: We do not understand why we should use a k-mean clustering to classify the data if our purpose is to predict the ΔGbind value. The models we have presented are continuous models not classifications models. We are wondering if reviewer may tell us how a classification technique may improve our regression models.
Authors should present the AD (applicability domain) of their data to ensure the good variation of their training and validation sets.
Answer: We agree with the reviewer that a clear representation of the AD is a key factor to assess the distribution of the training and validation subsets. Due to this, we have added the AD for both models (Figure 7 and 9).
Figure 7. Applicability domain
Figure 9. Applicability domain
Finally, we appreciate the efforts of the editorial board of Biology and reviewers for improve the quality and clarity of our manuscript.
Looking forward to hearing from you.
Yours sincerely,
Michael González-Durruthy.
Riccardo Concu

Reviewer 2 Report
Dear authors, this is very important and interesting topic. The first sentence in Abstract maked me interesting, the second – it seems to be very good work. However, next paragraphs & Figures, sorry, but how can you underline the SWCNT while yours experimental material are very large multiwalled CNTs (Fig.1)? Unfortunately there is no word about the source of the tubes.
The scale problem: SWNCT are at least 100x tinier then those presented if Fig.1. On the other hand are much longer than those presented in scheme in Figs 1&2.
On the other hand authors describe computational details, why? These are implemented in the software.
Finally biological problem: how to force CNT to be localized exactly in these cell’s regions. Moreover, the values of interaction energies (I) small – CNT, (II) kcal/mol per “mol” of what?
Thus, the work is unacceptable.
Author Response

(The authors gave the same response as above.)

Reviewer 3 Report
- The paper is so verbose that it becomes impossible to follow the discussion of the sole paper. It contains so much general information that could be moved to the supplementary or summarized without the loss of the integrity of the paper. Authors are encouraged to reduce the size of their paper to a maximum of 12-15 pages and move all general information to the SI.
- The processing of SWCNT is not clearly demonstrated. As the authors correctly mentioned the SWCNT tends to agglomerate owing to the high van-der-Waals interactions and without good dispersion, the discussion on the interaction of SWCNTs with other species is erroneous. TEM images on page 9 of the paper clearly demonstrated the heavily agglomerated area of SWCNTs which adversely affected their interaction with other moieties.
- It is not cleared if the authors do the experimental works by themselves or they just model some other work data. If they perform their own experiments, they should state the chemical and their grade which are using.
- Authors should apply more robust algorithms like k-mean clustering for the classification of their data.
- Authors should present the AD (applicability domain) of their data to ensure the good variation of their training and validation sets.
- The authors used SWNT-COOH for their modeling and it is not cleared how to interpret this structure. Normally oxidation in SWCNTs induced by acid-treatment and the outer surface of SWCNT will be covalently covered with COOH. In practice, the smaller SWCNTs oxidized more (contain more -COOH) compared to the larger ones. It is also not clear who the aspect ratio of SWCNTs affected their interactions.
Author Response

(The authors gave the same response as above.)

Round 2
Reviewer 1 Report
The authors have addressed all the comments, the manuscript can now be accepted for publication
Author Response
Dear reviewer,
please find enclosed a pdf with the response to your questions.

Reviewer 2 Report
Dear Authors unfortunately, you did not improve the work:
1. you need to prove you have SWCNT (till now you show MWCNT!)
2. you need to fully characterize to prove that it is the SWCNT.
3. you need to show that there -COOH exist.
Sorry, but the work still is unacceptable.
Author Response
Dear reviewer,
please find enclosed a pdf with a response to your points.
Kindly regards,
Dr. Riccardo Concu
Dr. Michel Gonzlaez Durruty

This manuscript is a resubmission of an earlier submission. The following is a list of the peer review reports and author responses from that submission.
Round 1
Reviewer 1 Report
The manuscript claims description of experimental and computer simulation studies of 'Carbon Nanotubes Nanotoxicity'. This area of investigations have attracted attention in previous decades after introduction of CNT into R&D. It was undoubtly established that at reasonable (not excessive) concentrations SWCNT do not posses toxic effects. The SWCNT concentration in ug/ml mentioned in the manuscript is huge and can not be realized in practice. Moeover SWCNT do not exist separately but only in bundles. For obtaining of individual nanotubes the bundles must be detroyed by special techniques assuming that each individual CNT will be covered by a layer hindering formation bundle aggregates of CNT. In view of this motivation of the work reported in the manuscript seems to be not suitable, proposed and used methodology (experimental and computer simulation) not reasonable, discussion and conclusion speculative.
Moreover the authors of the manuscript demonstrate no intention to do their work understandable.
Also most of cited works has no relation to the matter discussed. And legends to figures do not explain and even not mentioning in many cases information presented.
Reviewer 2 Report
This is an interesting article dealing with the evaluation of the effect of single-walled carbon nanotubes on mitochondrial F0F1-ATPase activity. Authors combined experimental and computational assays to assess the CNT nanotoxicity.
The manuscript is well organized and the experimental and computational set-up is well designed and suitable for obtaining the expected results which were found in agreement with the authors hypothesis.
It is an opinion of this reviewer that the manuscript can be published after minor revisions as follows:
- Can authors improve the introduction or discussion sections to better highlight the expected advantages of the proposed approach?
- Please check the author contribution section to better fit with author guidelines
Reviewer 3 Report
This is a highly computational manuscript that seeks to characterize the toxicological modulation of mitochondrial ATP bioenergetic mechanisms due to exposure to single-walled carbon nanotubes (SWCNTs). The modeling experiments examining the molecular docking interactions of SWCNTs with the mitochondrial F0-ATPase complex seem like they should be the highlight of this manuscript, but ultimately feel underdeveloped and buried by the rest of the manuscript. The conclusions drawn based on Figures 3 and 4 do not appear to be fully supported by the data (Fig 3A and Fig 3D are clearly not similar – why do they have identical FD(BW) values, and how can that indicate that they are similar when their only apparent similarity is that they are both symmetric about a diagonal axis?) and, because they lack a clear mechanistic explanation that would be of consequence to future research, appear to be incremental at best and already established at worst. This manuscript would be strengthened significantly if the data in Figures 3-5 were more clearly presented and the significance of those results in the context of past and future research were more clear. There are also concerns regarding the statistical analysis. It is unclear if the replicates of the experiments using rat-liver sub-mitochondrial particles (SMPs) are technical replicates or biological replicates (i.e., using SMPs from multiple rats). The following minor points could also be addressed to further strengthen this manuscript.
- It is unclear why both pristine and carboxyl-modified nanotubes are investigated until Figure 3. Including this information in both the introduction and the abstract would strengthen this proposal.
- It would be helpful to show an introductory figure of F0F1-ATPase with relevant parts labeled (e.g., i-sensor and j-effector residues, SWCNT docking site)
- No characterization or reference is provided to verify the isolation of F0F1-ATPase using the protocol in lines 110-123.
- The abbreviation RLM is first used on line 95, but not defined until line 127.
- Please define all terms in ΔGint in line 169.
- Please provide the value of the interaction constant γ on line 225.
- Please provide references with descriptions of the molecular descriptors used.
- Please clarify what the data in the graphs in Figure 1 represent (e.g., mean ±D.) and include negative error bars.
- It is unclear why three symbols are used (i.e., *, **, #) all for p<0.05.
- It is unclear how the data shown in Figure 3 relates to protein conformation.
- The terms FDB+BW and FDW+BW appear to be switched on lines 391 and 392.